# Conceptualizing an Integrative Multiple Myeloma Care: The Role of Nutrition, Supplements, and Complementary Modalities

**DOI:** 10.3390/nu16020237

**Published:** 2024-01-11

**Authors:** Francesca Andreazzoli, Ilana Levy Yurkovski, Eran Ben-Arye, Massimo Bonucci

**Affiliations:** 1Department of Hematology, Versilia’s Hospital, Viale Aurelia, 335, 55049 Camaiore, Italy; 2Hematology Unit, Bnai Zion Medical Center, Haifa 3339419, Israel; 3Rappaport Faculty of Medicine, Technion—Israel Institute of Technology, Haifa 3109601, Israel; eranben@netvision.net.il; 4Complementary and Integrative Medicine Service, Bnai Zion Medical Center, Haifa 3339419, Israel; 5Integrative Oncology Program, The Oncology Service, Lin Carmel, and Zebulun Medical Centers, Clalit Health Services, Haifa 3535152, Israel; 6Artoi Foundation, Via Ludovico Micara, 73, 00165 Rome, Italy; maxbonucci@artoi.it

**Keywords:** Multiple Myeloma, nutrition, acupuncture, vitamin D, supplementation, integrative medicine

## Abstract

Multiple Myeloma (MM) is the second most prevalent hematologic malignancy, and its incidence has been increasing enormously in recent years. The prognosis of MM has changed radically with the introduction of new drugs that have improved life expectancy; recurrences are a common occurrence during the course of the disease and are characterized by an increase in refractory to treatment. Moreover, MM patients are challenged by quality of life-related concerns while limited conventional therapy may be offered. This includes bone pain and dialysis due to the complications of acute renal failure. We, therefore, believe that it is very important to add new treatment modalities, including supplements, nutritional modifications, acupuncture, and mind–body therapies, with the goal of improving treatment tolerance, effectiveness, and patients’ quality of life. Moreover, many patients use some of these supplements on their own, in the hope of reducing the side effects, so it is even more important to know their action and potential. The purpose of this review is to illustrate all these strategies potentially available to enrich our approach to this, to date, incurable disease.

## 1. Introduction

Multiple Myeloma (MM) represents 10–15% of all hematological malignancies [1,2,3].

The last few years were characterized by a significant improvement in available therapies for this disease, including new-generation immune modulators and proteasome inhibitors (PIs), monoclonal antibodies, or immunotherapy (bispecific T-cell engagers or Chimeric Antigen Receptor T cell therapies, so-called CAR-T) [2]. 

Following this progress, the median overall survival has more than tripled [2]. However, MM is still an incurable disease and has become a chronic condition with patients experiencing many symptoms related to the disease itself or its treatment, which impacts health-related quality of life (QoL) [4,5,6,7]. These include pain, fatigue, sexual dysfunction, sleep disorders, immune deficiency, anxiety, depression, and/or loss of control [5,6,7,8], making patients with MM the most symptomatic in the haemato-oncologic field [9]. In the last few years, Integrative Oncology (IO) has been developed, and subsequently, Integrative Hematology (IH) was defined as a separate entity [10]. IO is a “patient-centered, evidence-informed field of cancer care that utilizes mind and body practices, natural products, and/or lifestyle modifications from different traditions alongside conventional cancer treatments, and aims to optimize health, QoL, and clinical outcomes across the cancer care continuum” [11].

Specifically, in the case of MM, patients have been shown to use complementary and alternative medicine widely but, generally, without informing their hematologist [12], potentially leading to safety issues [13]. The integrative approach to the treatment of myeloma therefore has many objectives: firstly, a reduction in disease-related symptoms such as bone pain, which is very detrimental to the patient’s life [9,14,15]; and a reduction in treatment side effects, allowing greater compliance to them as well as greater effectiveness. Another goal is to reduce the toxicity of drugs and improve the performance of the immune system. A patient with MM has many unmet needs, and a patient-centered approach, which includes mind–body techniques, can help to better manage the disease and therapies.

Finally, some techniques have also shown efficacy in improving the course of the disease [16]. In the present work, we aimed to review the different kinds of complementary and integrative techniques that may be effective and safe for the management of MM patients. The goal is also to propose preliminary consensus guidelines based on the existing literature for the safe and effective use of IH in MM.

## 2. Nutrition, Lifestyle Changes and Microbiota

### 2.1. Nutrition, Lifestyle Changes, and the Link with Pathogenesis and Progression of MM

Recently, a link between nutrition, lifestyle, and MM has emerged [17]. In addition, with the lengthening of the life of MM patients due to an improvement in the available treatments, it may be useful to design a nutritional-based consultation on how to improve the QoL of patients coping with MM. In particular, Monoclonal Gammopathy of Unknown Significance (MGUS) and Smoldering MM are situations in which preventive interventions may be considered. 

In a study conducted with patients with MM, following diagnosis, 82% reported nutrition-related questions addressed to non-medical sources rather than the hematologist. Furthermore, 94% of those who received advice from a hematologist followed their instructions, which indicates that the hematologist’s expertise can address this need [18].

Firstly, nutrition could play a role in MM prevention strategy. This could be important to take into account: the protective role in the pathogenesis of the MM of some foods may also be associated in some cases with an antitumor role in the stage of overt disease.

Several studies suggest an association between diet and MM incidence. For example, the Epic study shows cancer risk reduction in vegetarians, vegans, and fish eaters, especially regarding MM [19].

Some case-control studies confirm the trend toward the protective action of fish use [20,21]. A meta-analysis [22] focusing precisely on the relationship between fish intake and myeloma showed an inverse but not a linear relationship, suggesting that an intermediate weekly consumption, not too frequent, may be the right compromise between an intake of omega 3, presumably responsible for the beneficial effect, and that of ocean pollutants, unfortunately, present especially in large fish.

Omega 3, in addition to the known anti-inflammatory effect, has demonstrated a specific action on mouse models of MM in vitro in which they can reduce cell growth. The reduction in interleukin 6 (IL6) production could also be involved in specific anti-MM action [23].

Although some case-control studies have not shown a significant association between fruit consumption and MM risk, there is evidence showing a linear beneficial trend for such an association [24,25].

A recent Icelandic prospective study of cohorts showed that fruit consumption (equal to or greater than 3 times a week in adolescence) is associated with a reduced risk of MGUS and, when consumed in adulthood, with reduced progression of MGUS to MM [25].

Also, adequate consumption of cruciferous, tomatoes, and whole grains may reduce the risk of MGUS and MM, and an excess of refined sugars and sugar-sweetened beverages can increase it [25,26].

The beneficial effect of plant-based foods could be attributed to the action of dietary fibers on the production of butyrate-producing bacteria such as Faecalibacterium Prausnitzii. Also, the presence of molecules with antitumor activity such as isothiocyanates, indoles, and flavonoids (for example, lycopene from tomatoes) can be protective. Fibers and micronutrients in whole grains may be associated with the metabolic effects of glycemic index modulation, reduced insulin stimulation, and reduced production of insulin-like growth factor (IGF1). IGF1 is a molecule well known for its stimulating action on proliferative molecular pathways; also, in MM, its activity has been documented to involve the processes of homing, cell proliferation, apoptosis, angiogenesis, and bone metabolism [27]. The Epic study and other cohort studies show a higher incidence of MM in meat eaters and dairy consumers [19,28]. The high levels of IGF1 resulting from the consumption of dairy products could partly explain this correlation. *N*-nitroso compounds, heterocyclic amines, and heme present in meat could exert an oxidizing and pro-carcinogenic action as well.

Among other lifestyle factors, alcohol has a controversial impact on the incidence of MM, with a significant reduction in the incidence in moderate drinkers (2–2.9 drinks/day) [29].

It is not clear why this beneficial action of alcohol is different from what happens in other cancers. Still, it could be linked to a direct modulation of the immune system in light and moderate drinkers [30].

### 2.2. Obesity, Sarcopenia, and MM

The relationship between obesity, MGUS, and myeloma is well known. Not only does obesity affect the incidence of MM but also the progression from MGUS to MM and MM mortality rate. Many mechanisms are involved: the production of IL6 and leptin by fat cells leads to increased inflammation and cell proliferation; the reduction in adiponectin is associated with increased IL6, activation of NFkB, and osteoclastic maturation. Finally, the presence of a more significant adipose tissue in the bone marrow stimulates the proliferation of MM cells and their migration outside the bone marrow [31].

Another important risk factor to consider at diagnosis is the nutritional status of the patient, especially the presence of sarcopenia, which is a negative prognostic factor.

Sarcopenia is frequent in patients with MM at diagnosis; the presence of reduced subcutaneous adipose tissue is associated with a worse prognosis [32].

The Controlling Nutritional Status (CONUT) score is a simpler tool to use but equally effective in predicting the outcome of MM patients, especially those undergoing autologous stem cell transplantation (ASCT) [33]. 

Finally, the risk of muscle mass deficiency must never be neglected, and an adequate protein intake must be guaranteed to the patient at diagnosis and throughout the treatment, which can itself contribute to sarcopenia: nausea, mucositis, and lack of physical activity due to bone pain can, in fact, reduce the patient’s food intake.

A daily protein intake of at least 1.4 g/kg appears to be necessary to obtain or maintain adequate muscle mass according to a recent systematic review [34].

### 2.3. The Role of Microbiota in the Pathogenesis and Treatment of MM

The microbiota of patients with MM shows less alpha diversity at diagnosis than controls [35]. Although there are conflicting data on the association of the disease with the presence of butyrate-producing bacteria, a dysbiotic feature seems to be present in patients with MM [35,36], and a correlation between levels of Interleukin 17 (IL17) in the bone marrow of patients with MM and the composition of the microbiota is currently being explored (NCT05712967) [37]. This increase could be induced by foods rich in salt and the subsequent increase in nitrogen-recycling bacteria (e.g., Klebsiella and Streptococcus) [38].

The microbiota is also inevitably involved in the link between nutrition and MM mentioned above, being the first mediator of the effect of food on the body.

This could affect not only the pathogenesis of the disease but also its course and response to therapy. A clinical study of 34 MM patients in maintenance therapy with lenalidomide showed interesting results in this regard. Patients following a plant-based diet had a higher fecal concentration of butyrate, the consequence of an increase in butyrate-producers E. Hallii and F. Prausnitzii [39]. The increased fecal butyrate concentration was associated with a more sustained minimal residual disease negativity, which is known to be predictive of longer Progression-Free Survival [40].

Although this is a small study, these results suggest and confirm an essential association between nutrition, microbiota, and the effect of conventional therapy in MM patients. Recently, the detection of a specific microbiota in the bone marrow of patients with MM, mainly localized intracellularly, widens this scenario. The researchers hypothesize that the presence of intratumor MB is a consequence of an impairment of the gut MB and intestinal permeability, confirming a direct relationship between gut MB and the pathogenesis of MM [41]. Further well-designed clinical studies are necessary to clarify the therapeutic potential of an integrated approach that includes a “pro-microbiota” diet. At present, the Nutrivention3 study (NCT05640843) is a randomized, multi-center pilot study comparing 150 patients to evaluate the effects of a plant-based diet versus an omega 3 or placebo supplementation in patients with MGUS or SMM and high Body Mass Index on the concentration of fecal butyrate [42].

## 3. Natural Compounds and Vitamins: Their Potential Role in the Treatment of MM

### 3.1. Curcumin

Curcumin (diferuloylmethane), the main bioactive component of Turmeric (curcuma longa), has demonstrated anti-inflammatory, antibacterial, and anticancer in vitro effects against many cancer cell lines [43].

In MM, its action is expressed both by affecting specific signaling pathways and with an epigenetic modulatory effect.

Among the various effects, Nuclear factor kappa B (NFκB) inhibition appears to be particularly important in this disease: one of the backbones of MM therapy is, in fact, constituted by PIs, which cause NFκB inhibition. This would suggest a possible hypothetical interaction between curcumin and PIs.

In vitro, curcumin has been shown to enhance the cytotoxic effect of bortezomib (the first PI) [44]; it also synergizes with carfilzomib through a major downregulation of NFkB [45].

Curcumin was also reported to downregulate the expression of cyclin D1, inhibit STAT3 phosphorylation, and reduce IL6, even in patients with unfavorable cytogenetics and regardless of Tumor Protein 53 (TP53) status: this sensitivity to curcumin is also present both in the cells of patients at diagnosis and at recurrence, demonstrating that resistance phenomena do not occur [46,47].

Finally, it inhibits osteoclastogenesis by inhibiting RANKL [48].

The main reason for the incurability of MM could be related to the presence of cancer stem cells (CSCs): they confer treatment resistance and are associated with recurrence and poor prognosis. Recently, curcumin has been demonstrated to be efficient against these cells, increasing apoptosis [49].

As an epigenetic modulator, curcumin also appears to act in MM by inducing DNA methyltransferase (DNMT)-mediated mTOR methylation: hypermethylation of the mTOR promoter region would inhibit the antiapoptotic action of this molecular pathway [50]; unlike the hypomethylating drugs used in hematology, such as azacitidine, curcumin would not cause an overall change in DNA methylation but only the specific one caused by DNMT3a and DNMT3b, and this could give it more peculiar anti-myeloma activity.

Finally, in some mouse models, it increases chemosensitivity to dexamethasone, doxorubicin, and melphalan and synergizes not only with the action of bortezomib and carfilzomib but also with immunomodulatory imide drugs (IMiDs) such as lenalidomide and thalidomide [51]; synergy with lenalidomide is expressed through the suppression of the cereblon gene, which is exactly the target of the IMiDs. 

However, the literature on the association of these promising in vitro effects with effective clinical action is presently limited.

The first study, published in 2009, involved 26 patients with MGUS treated with 4 g daily of curcumin; in this study, the monoclonal component decreased in 50% of patients, in keeping with a reduction in markers of bone resorption [52]. Following, the authors published a randomized double-blind placebo-controlled trial in both MGUS and smoldering myeloma: they used curcumin in doses of 4 and 8 g daily, and this led to a reduction in the chain k/λ ratio by 35 and 36%, respectively, as well as the markers of bone resorption [53]. As not all cases of MGUS and smoldering myeloma evolve into frank disease, the use of conventional chemotherapy is not justified in these cases; precisely for this reason, supplements such as curcumin express all their chemopreventive potential in this phase of the pathogenesis of the disease.

In this regard, Zaidi et al. [54] published a case study in which curcumin (8 g daily) was associated with hyperbaric sessions that led to disease control in a case of refractory relapsed MM for 60 months, with an excellent safety profile.

A more recently published case study reported the efficacy of curcumin in replacing dexamethasone in patients who were intolerant to the latter. The authors treated 15 patients who were under treatment with regimens containing either IMiDs or PI combined with dexamethasone, substituting the latter with C3 complex curcumin at a dosage of 3–4 g daily. The researchers concluded that curcumin may act as a steroid-sparing agent in patients with MM who are intolerant of dexamethasone, with an excellent tolerability profile [55]. These data are exciting considering that in MM, we often use therapeutic regimens, if effective, up to progression; if these medications, as often happens, contain cortisone, the patient is frequently subjected to massive and prolonged doses of steroids and their heavy long-term side effects. Even with the aim of reducing steroid load and toxicity, curcumin is, therefore, an engaging weapon to take into consideration.

Finally, a pilot randomized trial was conducted [56] in which 33 newly diagnosed patients, ineligible for transplant, were randomized to receive melphalan 4 mg/m^2^ plus prednisone 40 mg/m^2^ for 7 days and curcumin 8 g/daily for 28 days or placebo for 4 cycles. Curcumin-treated patients showed a higher treatment response rate (75% vs. 33.3%) with a contextual reduction in NF-κB, IL-6, VEGF, and TNF-α levels. These few but promising results emphasize the need for more well-designed clinical studies to explore the potential additive healing effect of curcumin, as well as to identify the type of biochemical form most available (e.g., extract, liposomal, and water-soluble form).

### 3.2. Epigallocatechin Gallate (EGCG)

EGCG has shown an antiapoptotic effect on MM cells in vitro through the stimulation of ROS and a reduction in the levels of peroxiredoxin (an antioxidant molecule); moreover, a crucial role is played by the selective interaction with the laminin 1 receptor, which is much higher in patients with MM than in controls and whose absence prevents the apoptotic effect of EGCG [57]. Other mechanisms involve the inactivation of the enhancer of zeste homolog2 (EZH2) and the mitochondrial apoptosis pathway [58].

Although the in vitro antitumor efficacy of EGCG in MM is undisputed, its in vivo role and, in particular, its interaction with bortezomib, a backbone of the induction therapy, is controversial. In 2009, two contradictory studies were published in this regard. Golden et al., demonstrated that EGCG and other polyphenols derived from green tea prevented tumor cell apoptosis induced by bortezomib in vitro and in vivo at concentrations also easily achievable in humans: the antagonistic action of EGCG was evident only with boronic acid-based PIs, so the binding with the boronic acid of the molecule appears to be the mechanism underlying this inhibitory mechanism [59]. In the second work, however, a synergistic effect between the two substances was demonstrated, and this would be due, in the opinion of the authors, to the higher dosage of EGCG and bortezomib used [60]. The discrepancy of these data, the lack of clinical studies, and the possible difference between what happens in vitro and what occurs in vivo due to certain variables such as absorption and bioavailability impose a prudent attitude, especially until well-designed clinical trials are available.

Some more recent studies affirmed again the antagonism between the two substances [61,62]; in particular, Qiu et al. [62] have shown that EGCG may neutralize bortezomib-induced apoptosis, activate Wnt/β-catenin signaling, and result in the accumulation of beta catenin, which subsequently activated c-myc and cyclin D1. The authors conclude by discouraging green tea intake during bortezomib therapy.

Finally, EGCG is very interesting for its action on glutaminolysis: MM cells have a glutamine addiction, which is responsible for their dependence on glutamine uptake. In a recent study [63], Li et al., showed that the combination of EGCG and telaglenastat, a glutaminase inhibitor, synergistically inhibits proliferation and induces apoptosis in MM cells in vitro. Targeting glutamine metabolism, therefore, should be a winning strategy.

### 3.3. Vitamin D

Vitamin D deficiency is found in most patients with MM at diagnosis, as in many other types of cancer. This deficit is described as “alarming” in patients with bone involvement [64]. Even some peculiar polymorphisms of the vitamin D receptor (VDR) gene may be a molecular marker of the risk of development of the disease [65].

Low levels of vitamin D correlate with an increased C-reactive protein (CRP) and the ISS stage at the time of diagnosis: they may, therefore, be predictive of more advanced disease [66]. It is very interesting to observe how the action of some drugs used in MM also passes through the modulation of the molecular pathway of vitamin D: bortezomib, for example, upregulates the production of VDR, and this effect is amplified in the presence of vitamin D.

Thalidomide- or bortezomib-induced neuropathy is more severe in the case of concomitant vitamin D deficiency, which, however, does not appear to influence the incidence [67].

Vitamin D and its active metabolite, calcitriol, act as modulators of the immune system: absolute lymphocyte count recovery and relapse-free survival after ASCT are improved after daily administration of calcitriol of 0.25 μg, as shown in a randomized-controlled trial (RCT) vs. placebo [68].

Interestingly, vitamin D can increase CD38 expression in plasma cells, thereby amplifying the binding of CD38-targeting antibodies; however, the cytotoxicity of the latter could be compromised in case of vitamin D deficiency, which would cause a worse performance of the macrophages associated with myeloma (MAMs). Lenalidomide and pomalidomide can restore the vitamin D pathway in the MAMs, improving their cytotoxicity. In doing so, vitamin D becomes a crucial link to improve the synergy between immune modulators and monoclonal antibodies [69].

The above data may support the rationale to test vitamin D levels at diagnosis and supplement them before starting therapy and during treatment with anti-CD38 and immunomodulatory drugs. 

Noteworthily, vitamin D levels do not appear to be significantly associated with the coexistence of bone disease [70]. 

In Figure 1, the main mechanisms of action of the supplements most studied in the MM are presented.

### 3.4. Others

Ascorbic acid can reduce Bortezomib-induced neurotoxicity by leading to the recovery of damaged Schwann cells; unfortunately, it has been shown to inhibit Bortezomib cytotoxicity in vitro and in vivo, so its supplementation in patients who are on treatment with this PI is not recommended [71,72].

Despite its importance for the immune cells, it is ineffective in improving bone marrow recovery after ASCT both in MM and lymphoma patients, so its use is not advised in this setting [73].

Sulforaphane is an isothiocyanate derived from cruciferous vegetables, which has been shown to have similar action to bortezomib in inhibiting the degradation of inhibitory kappa B kinases (IkB), which, in turn, inhibits the proteasome complex.

Moreover, it shows synergy with dexamethasone, doxorubicin, bortezomib, and melphalan. In in vivo models, it leads to a reduction in the burden of disease and an increase in survival [74].

Resveratrol has shown strong antiangiogenic activity by regulating factors such as VEGF, matrix metalloproteinase-2 (MMP2), and matrix metalloproteinase-9 (MMP9).

By constitutively activating STAT3 and NFκB, it overcomes an essential mechanism of chemoresistance and it has a synergistic interaction with Bortezomib and Thalidomide [75]. There is insufficient clinical evidence to understand the clinical efficacy of resveratrol. In a single phase II study, it was administered in association with bortezomib: in this clinical experience, the toxicity profile was unacceptable, with 50% of patients developing severe renal failure. To date, resveratrol in combination with bortezomib is strongly discouraged. Further studies are needed to clarify the safety of this natural compound in this setting of patients [76].

Cannabinoids are compounds found in plants from the genus Cannabis, of which the most widely used is Cannabis Sativa. In recent years, their role in managing the patient’s symptoms has emerged and they are used primarily to reduce chronic pain and chemotherapy-induced nausea and vomiting and stimulate appetite [77]. Since many different cell lines express CB1/CB2 receptors for cannabinoids, their anti-cancer potential has been extensively investigated recently, in in vitro and in vivo models: among the various plant components most studied for their antiapoptotic activity are delta-trans-9-tetrahydrocannabinol (THC) and cannabidiol (CBD), with the latter being free of psychoactive effects. In contrast to the in vivo and in vitro settings, clinical research is limited, though suggests good patient tolerability even at high doses, particularly in glioblastoma and MM [78]. In MM, in vitro studies have demonstrated reduced viability of myeloma cells and proapoptotic activity in cannabinoid-treated MM cells, including dexamethasone-resistant cells. Cannabinoids can reduce the expression of the β5i subunit of the immunoproteasome, therefore increasing the efficacy of carfilzomib, a second-generation PI. In addition, they allow resistance mechanisms effectively linked to the expression levels of β5i to overcome [79]. These compounds are, therefore, promising not only for improving the symptoms and QoL of patients, but also for their anticancer and synergistic activities with currently used drugs. Further studies are needed to confirm this promising action.

Table 1 shows the potential use of the substances described above.

## 4. Acupuncture

Pain is a common QoL-related concern experienced by MM-diagnosed patients, with various etiologies, such as mechanical due to bone lesions, neuropathic due to the side effects of prescribed drugs (mainly PIs or immune modulators), infections (post-herpetic neuralgia), or direct infiltration of light chains or amyloid into peripheral nerves [80]. The management of these symptoms with conventional drugs is generally unsatisfactory and subject to side effects, while acupuncture has been shown to relieve cancer-related pain safely [81]. In a bibliometric analysis summarizing research conducted between 2012 and 2022, acupuncture was suggested as a leading modality in the treatment of MM-related pain [82,83]. This is particularly in regard to treatment of chemotherapy-induced peripheral neuropathy (CIPN), which is an unmet need in Western medicine [80]. Acupuncture has been widely studied for this condition with positive results [84,85]. Specifically, the widely used proteasome inhibitor (PI) bortezomib can cause CIPN in up to 40% of patients [86]. A number of studies have demonstrated the effectiveness of acupuncture for bortezomib-induced peripheral neuropathy (BIPN). Following case reports [87], a retrospective case series on 5 patients demonstrated the immediate and long-lasting pain reduction associated with acupuncture in BIPN [88]. Acupoints included bilateral ear points and body points (Large Intestine 4 (LI4), Triple Energizer 5 (TE5), Large Intestine 11 (LI11), Stomach 40 (ST40), and BaFeng) on a weekly basis. The effect of the same protocol has been confirmed in 27 patients with BIPN, in which a reduction in specific symptoms has been observed (numbness/tingling in hands and feet, cold sensitivity, and unpleasant feelings), although nerve conduction velocities (NCV) were unchanged [89,90]. Electroacupuncture has also been shown to reduce pain severity and improve QoL, as well as specific motor functions in 19 patients with grade 2 or higher thalidomide- or bortezomib-induced neuropathy [91]. In a 54-patient Chinese RCT, the combination of acupuncture + moxibustion was shown to improve QoL compared to conventional drugs, although no difference was shown in the improvement of NCV [92]. Finally, an additional RCT assigned 104 myeloma patients with CIPN to a methylcobalamin-alone group or one associated with three 28-day cycles of acupuncture in acupoints different than previously described (LR3, ST43, GB41, SP6, ST36, SP10, ST25, GV14, GV12, GV11, GV9, BL13, BL17, and BL58). Each treatment cycle included three daily acupuncture treatments followed by 10 days of treatment once every other day.

This study showed that adding acupuncture led to a significant pain reduction while improving both daily activity and nerve conduction velocities compared to methylcobalamin alone [93]. These studies suggest that acupuncture may be an effective treatment to improve QoL in patients suffering from Myeloma-related CIPN, although its effect on nerve conduction is controversial.

Another important challenge in patients with MM is ASCT, which is still a common practice guideline in fit patients following first-line induction and a reduction in disease burden. Deng et al., published an RCT with 60 patients who were assigned to true or sham acupuncture once daily for 5 days after receiving high-dose chemotherapy for ASCT [94]. Acupoints included GV20, Yintang (Ex-HN3), Heart 7 (HT7), Pericardium 6 (PC6), Stomach 36 (ST36), Spleen 6 (SP6), Kidney 3 (KI3), Liver 3 (LR3), and Ear Shen Men. Although the overall MD Anderson Symptom Inventory (MDASI) improved non-significantly during transplantation, at 15 days and at 30 days [94], true acupuncture was more effective than sham in alleviating nausea, poor appetite, and drowsiness [94,95]. This specifically improved sleep efficiency [96] and reduced the use of analgesics during ASCT and the number of opioid users post-ASCT among opioid-naïve patients [95,97].

Acupuncture’s mechanism of action for such symptom relief in MM patients has been studied as well. A Chinese RCT showed that the improvement of QoL in MM patients following heat-sensitive moxibustion was associated with decreased inflammatory markers, strengthened T-cell immunity, and a rise in the expression of some micro-RNAs (miR-125a, miR-140-5p, and miR-302′) [98]. These chemo-biological changes seem to be involved in the pathophysiology of acupuncture treating MM-associated symptoms. Finally, a murine model mice study demonstrated the synergism between acupuncture and bortezomib in improving survival in mice with MM. The pathophysiology seems to function by decreasing ornithine [99]. 

Last but not least, acupuncture-related safety concerns should be recognized in patients with MM and decreased immune system activity. This includes neutropenia and hypo-gammaglobulinemia exposing patients to severe infections, as well as thrombocytopenia during ASCT with its high risk of bleeding. The risk of infection following acupuncture therapy is very low. Still, we should note a recent case report on the exacerbation of pyoderma gangrenosum in a patient with MM treated with acupuncture [100]. Therefore, acupuncture should always be practiced with extreme precaution in MM and other immune-compromised patients with active infection, and for skin infection, one should avoid puncturing affected areas. As for severe thrombocytopenia, a recent retrospective chart review analyzing 815 acupuncture sessions on thrombocytopenic patients with hematological malignancies demonstrated the safety of acupuncture even with platelet counts below 20 × 10^9^/L [101]. Table 2 shows the possible indications of acupuncture in MM.

## 5. Mind–Body Medicine

Mind–body medicine is one of the main IO modalities encompassing a wide spectrum of interventions from traditional medicine systems like Tibetan, Ayurvedic (e.g., Yoga), and Chinese (e.g., Qi Gong) medicine to more contemporary Western approaches of relaxation, guided imagery, hypnosis, and mindfulness-based interventions (MBI). The hallmarks of many mind–body interventions are mindfulness and intention to berating, gentle touch, movement, and spirituality. Recent IO-related clinical guidelines on pain, depression, and anxiety co-published by the Society for Integrative Oncology (SIO) and the American Society of Clinical Oncology (ASCO) recommend mind–body medicine for the following indications: hypnosis in patients who experience procedural pain; MBIs, yoga, relaxation, and music therapy for treating symptoms of anxiety during active treatment; MBIs, yoga, tai chi, and/or qigong for treating anxiety symptoms after cancer treatment; MBIs, yoga, music therapy, and relaxation for depression symptoms during oncology treatment; and MBIs, yoga, and tai chi and/or qigong post-treatment depression [102,103]. In the field of Multiple Myeloma, Lamers et al., reported that as many as 52% of patients desired psychosocial interventions near diagnosis, where relaxation techniques were the most common preferred modality reported by 21% of patients [104]. This inclination toward mind–body intervention is perceived in the context of a coping strategy, which may also correlate with the immunological response after ASCT [105]. In the U.S., LeBlanc et al., reported a pilot RCT with a group of patients diagnosed with MM or Chronic Lymphocytic Leukemia, intending to examine the impact of a Mobile App to support self-management, which also included mindfulness activities [106]. The authors reported that distress tracking and guided meditations were among the most used functions within the app. In another study recruiting patients with a variety of hematological oncology diagnoses (13.8% with MM), patients were randomized to a single session of 30 min mindful breathing versus standard care and reported significantly less fatigue in the intervention group [107]. Bates and colleagues explored the impact of music therapy considered by some scholars in the context of mind–body medicine, while others categorize it independently among the art therapies discipline. The RCT explored symptom management following ASCT in patients with lymphoma or MM and found that the patients receiving music therapy used significantly less narcotic pain medication [108]. In the United Kingdom, semi-structured qualitative interviews in a purposive sample of MM patients suggested that spirituality is one of the main themes that patients perceive as important to consider regarding their QoL [7]. In a stimulating article, Saha and Mallik challenged rehabilitation with patients undergoing MM treatment, emphasizing patients’ unmet spiritual needs in India [109]. In this regard, it is of utmost importance to consider the cross-cultural context of spirituality, which may be regarded in different cultures in close proximity to religion and/or mind–body medicine. In this regard, Sherman and colleagues in the United States studied prospectively religious coping and orientation among 94 myeloma patients undergoing ASCT [110]. The results suggested that religious struggle may contribute to adverse changes in health outcomes in this clinical setting.

## 6. Discussion

This narrative review aimed to identify the tools that can integrate conventional MM treatment to enhance its effectiveness and tolerability, as well as the quality of life and overall survival of patients with MM.

We have identified four fundamental pillars to build this approach: nutrition, acupuncture, mind–body techniques, and the use of supplements.

Nutrition, also through the remodeling of the microbiota, is, in our opinion and based on the available scientific literature, not only a useful tool in the preventive stage or in cases of MGUS but also an essential therapeutic tool to meet patients’ unmet needs.

Many mechanisms are involved in food’s effects, such as the modulation of IGF1, the reduction in inflammation, and butyrate’s influence on the bone marrow microenvironment. This justifies the need for nutritional counseling and further clinical studies to lay the foundations for the definition of shared nutritional guidelines.

Acupuncture and mind–body techniques, which have a good safety profile, can provide valuable help in improving tolerance and compliance with treatment.

Among supplements and vitamins, we have identified those that have been most studied in MM both for their in vitro action and for their efficacy demonstrated in clinical studies. Most of the data we have available unfortunately come from preclinical studies that, although promising, are not yet confirmed by clinical trials. For this reason, well-designed clinical trials are absolutely necessary: they will allow us to propose guidelines that include the recommended supplements but also provide warnings for those with bad safety profiles. It is also very important to take into account the potential interactions with conventional drugs used in MM. As we have seen, some substances such as vitamin C may limit the effectiveness of drugs such as bortezomib. These are substances often used by the patients without communicating to their hematologist, thinking that they are harmless. In-depth knowledge of the mechanisms of action and possible drug interactions is therefore mandatory. These interactions could also be synergistic, opening up new horizons for integrative therapy, which could be a way of making the best use of the drugs currently at our disposal.

## 7. Conclusions

Integrative oncology modalities may offer significant improvement of QoL-related concerns in patients coping with Multiple Myeloma diagnosis and treatment. This is particularly relevant in the present era where effective hematology treatment is associated with increased life expectancy. 

In this narrative review, we examine the data regarding acupuncture, dietary interventions, and supplements, while also considering mind–body interventions as part of the entire hematologic approach (Figure 2).

More studies, primarily randomized controlled, are needed to explore if the effect of these IH modalities is limited to enhanced supportive and palliative care or may also contribute to better therapy efficacy. Not least important is exploring IH modalities’ effects regarding the safety and risk of drug interactions.

Further well-designed clinical trials will allow the construction of evidence-based guidelines.

## Figures and Tables

**Figure 1 nutrients-16-00237-f001:**
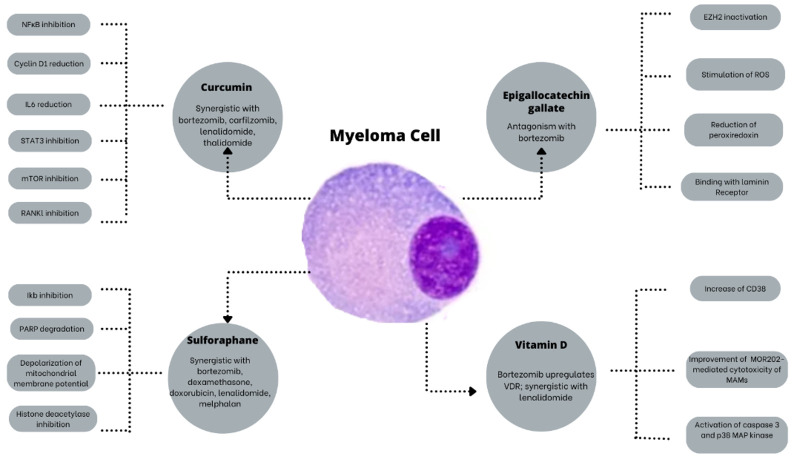
The main mechanisms of action of the most studied supplements in MM. EZH2: Enhancer of zeste homolog2; IkB: NkB inhibitor; IL6: interleukin 6; MAMs: Macrophages associated with myeloma; mTOR: Mammalian target of rapamycin; NfkB: Nuclear factor kappa B; PARP: poly ADP ribose polymerase; RANKL: Receptor activator of nuclear factor kappa-Β ligand; ROS: Reactive oxygen species; STAT3: Signal transducer and activator of transcription 3; VDR: Vitamin D receptor.

**Figure 2 nutrients-16-00237-f002:**
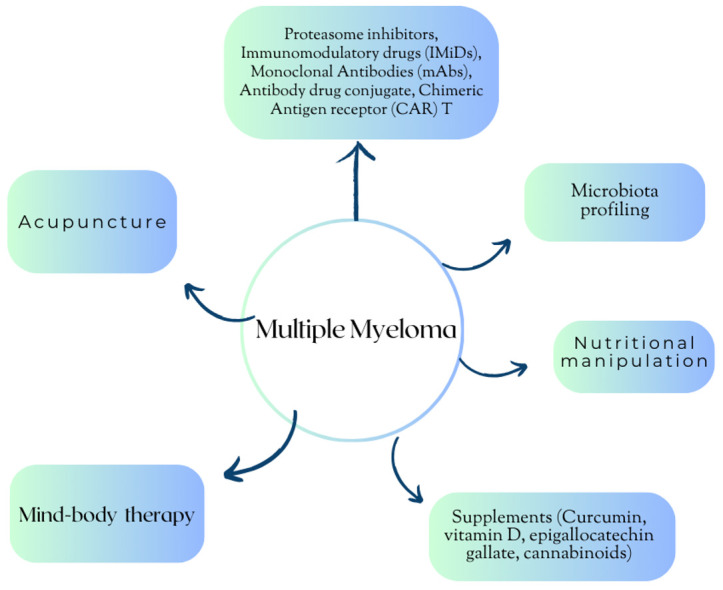
Multiple Myeloma: an integrative approach.

**Table 1 nutrients-16-00237-t001:** Potential use of supplements or natural compounds in Multiple Myeloma.

Substance	Dose	Potential Chemotherapy Interaction	References
Vitamin D	To reach blood levels >30 ng/ml	Synergism with lenalidomide, pomalidomide, anti-CD38	[67,68,69]
Curcumin	6–8 g daily or liposomal formulation	Synergism with bortezomib, carfilzomib, thalidomide, melphalan	[44,45,51,52,53,54,55,56]
Epigallocatechin 3 gallate	1 g daily	Antagonism with bortezomib	[61,62]
Vitamin C	6–8 g daily	Antagonism with bortezomib	[71,72]
Cannabinoids	?	Synergism with carfilzomib	[79]
Sulforaphane	To be found	Synergism with dexamethasone, doxorubicin, bortezomib, and melphalan	[74]
Resveratrol	To be found	Antagonism with bortezomib; possible side effects, not recommended	[75,76]

**Table 2 nutrients-16-00237-t002:** Proposed guidelines on the indications of acupuncture in Multiple Myeloma.

Indication	Description	Outcomes	Safety	References
CIPN	Ear and/or body points at least 1/week Electro Moxibustion	Improves sensory and motor symptoms. Improves QoL. Effect on NCV?	Avoid puncturing infected areas [37]. Safe for thrombo-cytopenia [38]	[87,88,89,90,91,92,93]
HSCT after conditioning regimen	1/day for 5 days in ear and body points	Alleviates nausea. Improves appetite. Improves sleep quality. Decreases pain. Reduces opioid use.		[94,95,96,97]

Legend: CIPN: chemotherapy-induced peripheral neuropathy; Electro: electroacupuncture; HSCT: hematopoietic stem cell transplantation; NCV: nerve conduction velocity; QoL: quality-of-life.

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
