# Peer review of "Conceptualizing an Integrative Multiple Myeloma Care: The Role of Nutrition, Supplements, and Complementary Modalities"

_nutrients, 2024, doi:10.3390/nu16020237_

Round 1
Reviewer 1 Report
Comments and Suggestions for Authors
The authors reviewed the association of nutrition, supplements, natural compounds, acupuncture, and mind-body medicine. I have some comments.
1. From line57-158, the authors need small sections for readability. Furthermore, the authors described the risk of MM development by fish intake, fruit consumption, etc. Considering the title of the manuscript, are nutrition and lifestyle before MM development needed? They are a little beside the title, and introduction. If you need, please make approproate section to read easily.
2. line158, "Natural compounds and vitamins" is difficult to understand. "Natural compounds and vitamins which effect pathogenesis and treatment of MM" ,etc more readable title is recommended.
Author Response
1. We divided the paragraph in subparagraph, as suggested, for better readability. We also explained why we talk about the link between nutrition and prevention, as rightly observed, adding this sentence: "Firstly, nutrition could play a role in MM prevention strategy. This could be important to take into account: the protective role in the pathogenesis of the MM of some foods may also associated in some cases with an antitumor role in the stage of overt disease."
2. We have modified the title as suggested
Reviewer 2 Report
Comments and Suggestions for Authors
The authors here provide a summary of potential non-drug modality as the part of integrative multiple myeloma care. It will stimulate the interest of discussion about this kind of concept in MM therapy. We think this paper can not be published in its present form unless my concerns are really solved.
1) There are many short paragraphs, making the whole structure not logically arranged.
2) The necessity and reason of integrative multiple myeloma care are not elucidated clearly. It is a very important point to be clearly presented.
3) All the possible factors should be listed in a summary table, though Table 1 is provided but not complete.
4) The potential working mechanism of the seven examplified nutrients in the integrative multiple myeloma care should be discussed and also provide an illustrating figure if necessary.
5) The conclusion part is too long and should be kept in brief.
6) In line 24, the word "side effect" is not suitable, maybe "additive effect" or "synergistic effect"?
7) Why are the nutrients in 3.1-3.7 not fully listed in Table 1? 8) The text in Figure 1 is not clear. 9) The potential relationship between these non-drug approaches and drugs in MM therapy should be also summarized in a separate paragraph. It is an important aspect to exhibit the significance of using other non-drug approaches in combination with effective drugs for gentle MM therapy.
Author Response
|
Comments 1: There are many short paragraphs, making the whole structure not logically arranged. |
|
Response 1: We incorporated the shorter paragraphs in a single entitled “Others”, to make the structure more uniform, as rightly observed |
Comments 2: The necessity and reason of integrative multiple myeloma care are not elucidated clearly. It is a very important point to be clearly presented.
Response 2: We added line 50-57 to better explain the aim of an integrative approach in MM
Comments 3: All the possible factors should be listed in a summary table, though Table 1 is provided but not complete.
Response 3: We added to the table the missing supplements as suggested
Comments 4: The potential working mechanism of the seven examplified nutrients in the integrative multiple myeloma care should be discussed and also provide an illustrating figure if necessary.
Response 4: we created a figure (name Figure 1) to explain better the working mechanism of the main supplements, focusing on those of which there is more data on the mechanisms of action in the pathological plasma cell
Comments 5: The conclusion part is too long and should be kept in brief.
Response 5: We shortened the conclusione as suggested
Comments 6: In line 24, the word "side effect" is not suitable, maybe "additive effect" or "synergistic effect"?
Response 6: We corrected the sentence as follows: "reducing side effects"
Comments 7: Why are the nutrients in 3.1-3.7 not fully listed in Table 1?
Response 7: At first, we opted not to include sulforaphane and resveratrol because of the poor clinical data in multiple myeloma
Comments 8: The text in Figure 1 is not clear.
Response 8: We modified the picture so that it is more readable
Comments 9: The potential relationship between these non-drug approaches and drugs in MM therapy should be also summarized in a separate paragraph. It is an important aspect to exhibit the significance of using other non-drug approaches in combination with effective drugs for gentle MM therapy.
Response 9: We tried to summarize the suggested concept in a new paragraph (number 6) titled “Discussion”

Round 2
Reviewer 2 Report
Comments and Suggestions for Authors
The authors have addressed all my concerns. The revised version has been much improved to be legible for all readers. It can be accepted in its present form. Congratulations to all authors!